# Relief of excited-state antiaromaticity enables the smallest red emitter

Heechan Kim [1,3], Woojin Park [2,3], Younghun Kim [1], Michael Filatov [2✉], Cheol Ho Choi [2✉] & Dongwhan Lee [1✉]

It is commonly accepted that a large π-conjugated system is necessary to realize low-energy electronic transitions. Contrary to this prevailing notion, we present a new class of light-emitters utilizing a simple benzene core. Among different isomeric forms of diacetylphenylenediamine (DAPA), *o*- and *p*-DAPA are fluorescent, whereas *m*-DAPA is not. Remarkably, *p*-DAPA is the lightest (FW = 192) molecule displaying red emission. A systematic modification of the DAPA system allows the construction of a library of emitters covering the entire visible color spectrum. Theoretical analysis shows that their large Stokes shifts originate from the relief of excited-state antiaromaticity, rather than the typically assumed intramolecular charge transfer or proton transfer. A delicate interplay of the excited-state antiaromaticity and hydrogen bonding defines the photophysics of this new class of single benzene fluorophores. The formulated molecular design rules suggest that an extended π-conjugation is no longer a prerequisite for a long-wavelength light emission.

[1] Department of Chemistry, Seoul National University, Seoul, Korea. [2] Department of Chemistry, Kyungpook National University, Daegu, Korea. [3] These authors contributed equally: Heechan Kim, Woojin Park. ✉email: mike.filatov@gmail.com; cchoi@knu.ac.kr; dongwhan@snu.ac.kr

Molecular light-emitters are finding wide applications from electronic displays to biological imaging[1–4]. When developing an optimal fluorophore for a given task, primary considerations are made to the size, wavelength, quantum yield, and synthetic tunability[5–10]. Within this context, low molecular weight fluorophores are recently gaining significant interest, as they can easily permeate cells with minimal perturbation of the biological system[11,12]. In solid-state device settings, small molecules also suffer less from intermolecular interactions and electronic coupling that often lead to luminescence quenching[13,14]. A large Stokes shift is another crucial factor in fluorophore design to reduce the inner-filter effect by minimizing spectral overlap between absorption and emission[5,6,15].

Existing strategies to realize large Stokes shift rely on (i) intramolecular charge transfer (ICT)[5,6,16], (ii) excited-state intramolecular proton transfer (ESIPT)[17–19], (iii) fluorescence resonance energy transfer (FRET)[20,21], (iv) desymmetrization of the molecular structure[6,22–24], (v) excimer/exciplex emission[25], or (vi) excited-state planarization of the benzannelated $8\pi$ system[26–28]. These design principles are applicable to known fluorogenic motifs, but often compromise other photophysical properties such as emission quantum yield. The development of small fluorophores having large Stokes shifts still remains an unsolved and challenging problem.

To build a minimal fluorophore, the $\pi$-delocalized benzene is a good starting point. However, the HOMO–LUMO gap of benzene is prohibitively large for practical applications (Fig. 1a). By introducing appropriate electronic controller groups, the frontier molecular orbitals (FMOs) of the parent benzene can be modulated (Fig. 1a) to bring the excitation and emission energy windows to the visible wavelength range. Such single benzene fluorophores (SBFs) have recently gained significant attention due to their unique photophysical properties of strong solid-state emission with large Stokes shifts (3000–8000 cm$^{-1}$)[29–35]. While the electronic origin of the large Stokes shifts of SBFs has often been ascribed to the HOMO–LUMO asymmetry implying ICT-type transitions[29–32], it is less clear how the small benzene core can promote significant charge separation.

In this paper, we disclose the chemistry of a new class of SBF, diacetylphenylenediamines (DAPAs, Fig. 1b). Built upon the archetypal aromatic benzene core with tight intramolecular hydrogen bonds, these molecules feature extraordinarily low molecular weight and large Stokes shift, as well as a wide spectral window that is tunable by facile and straightforward synthetic modifications. As a bonus, the small size and conformational rigidity of DAPA and its derivatives are ideally suited for in-depth theoretical and computational studies, which have been carried out for the first time for SBFs, including their excited-state dynamics of immediate relevance to de-excitation mechanisms. We found that the excited-state antiaromaticity of the benzene core itself[36,37], rather than the typically assumed ICT or ESIPT, is responsible for their peculiar photophysical properties.

## Results

**Serendipitous discovery and targeted synthesis.** Our entry into the chemistry of DAPA was aided by an unexpected discovery of the acid-catalyzed hydration–desilylation reaction of 3,6-bis((trimethylsilyl)ethynyl)benzene-1,2-diamine (**1**; Fig. 2). A quick literature search revealed that *o*-DAPA has been neither synthesized nor isolated, which is somewhat surprising given its simple structure. Visual observation of its green emission under UV lamp prompted our interest in its improved synthesis and comparative studies with regioisomeric *m*-DAPA and *p*-DAPA.

The three isomers of DAPA were independently synthesized by acid-catalyzed hydration–desilylation reactions of the corresponding TMS-protected diethynylbenzenediamines **1** and **3**, for *o*- and *p*-DAPA, respectively. For *m*-DAPA, the acetamide form **2** was used instead (Fig. 2). We found that *p*-toluenesulfonic acid works better as an acid catalyst. Either HCl or HBr promotes undesired hydrohalogenation reactions to reduce the yield. Each of these reactions was performed similarly to furnish *o*-DAPA in 45%, *m*-DAPA in 70%, and *p*-DAPA in 63% yields. Single-crystal X-ray crystallographic studies revealed the presence of tight intramolecular N–H···O hydrogen bonds with shortest N···O distances of 2.654(2) Å for *o*-DAPA, 2.686(2) Å for *m*-DAPA, and 2.701(3) Å for *p*-DAPA, between the amine and the acetyl groups (Fig. 2).

**Structure-dependent photophysical properties.** During the initial investigation, we realized that the photophysical properties of DAPAs are markedly different depending on the relative positioning of the hydrogen-bonding donor (HBD) and acceptor (HBA) pairs. As shown in the UV–vis absorption and fluorescence emission spectra (Fig. 3), CHCl$_3$ solution samples of *o*-, *m*-, and *p*-DAPA display characteristic $\pi \rightarrow \pi^*$ transitions at $\lambda_{\text{max,abs}} = 432$, 350, and 482 nm, respectively. Interestingly, both *o*-DAPA and *p*-DAPA show long-wavelength fluorescence at $\lambda_{\text{max,em}} = 531$ and 618 nm, respectively, whereas *m*-DAPA remains completely non-emissive.

While the two emissive isomers *o*- and *p*-DAPA have moderate quantum yields of 26% and 6%, they both display unusually large Stokes shifts of 4320 and 4570 cm$^{-1}$, respectively. As shown in the inset of Fig. 3, the emission spectrum of each isomer nicely mirrors the absorption spectrum, suggesting that neither large structural changes by ESIPT nor charge redistribution of

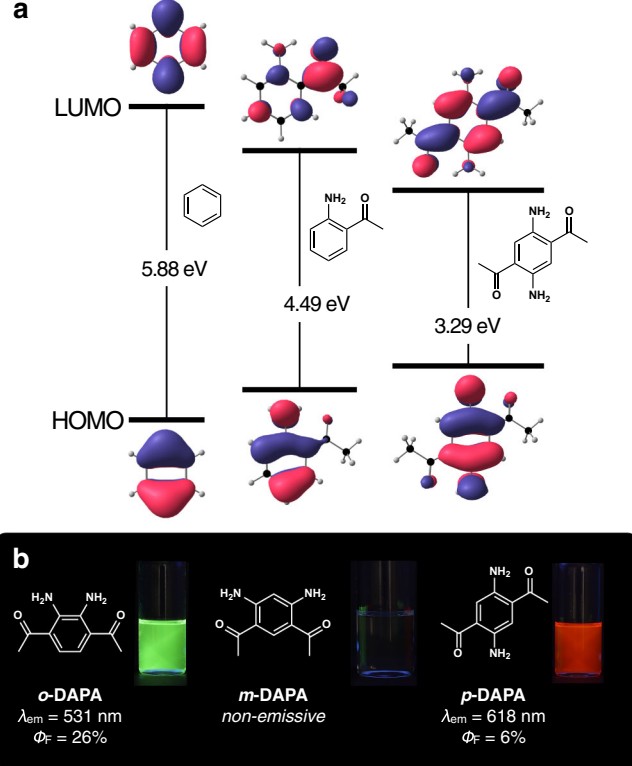

**Fig. 1 Single benzene fluorophores. a** HOMO–LUMO energy level diagrams of benzene derivatives calculated at the MRSF/BH&HLYP/6-31 G* level of theory. **b** Chemical structures of DAPA isomers along with photographic images of CHCl$_3$ solution samples taken under 365 nm UV lamp. LUMO lowest unoccupied molecular orbital, HOMO highest occupied molecular orbital.

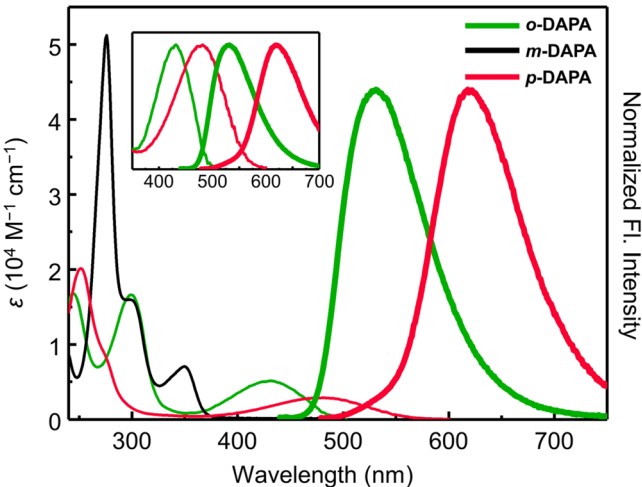

**Fig. 2 Construction of isomeric single benzene fluorophores.** Synthetic routes for *o*-, *m*-, and *p*-DAPA. *p*-TsOH *para*-toluenesulfonic acid, EtOH ethanol.

**Fig. 3 Structure-dependent light absorption and emission.** Absorption (thin lines) and normalized emission (thick lines) spectra of *o*-DAPA (green), *m*-DAPA (black), and *p*-DAPA (red) in CHCl₃ (sample concentrations = 50 μM). The inset compares the absorption (thin lines) and emission (thick lines) spectra of *o*- and *p*-DAPA; the absorption spectra are normalized to the absorbance at the longest maximum absorption wavelengths ($\lambda_{max,abs}$), whereas the emission spectra are normalized to the maximum fluorescence intensity. Fl., fluorescence.

ICT-type excited-state is involved in the experimentally observed radiative process (vide infra). We also noted that the fluorescence of *p*-DAPA is quite remarkable, since it represents the lightest molecule (FW = 192) displaying red emission (Supplementary Fig. 2).

To understand the electronic origin of such a large Stokes shift for deceptively simple-looking benzene derivatives, we formulated three key questions to be addressed. (i) How can *o*- and *p*-DAPA exhibit long-wavelength visible emission? (ii) What structural feature of *m*-DAPA makes the molecule non-emissive? (iii) Can

we structurally elaborate the minimalist DAPA fluorophore motif to cover the entire visible wavelength region?

**Structural modifications toward full-color fluorophores**. To test the utility of DAPA as a general SBF platform, an efficient synthetic protocol is needed to facilitate its structural diversification. The reaction of *o*-DAPA with carbonyl electrophiles, however, produces benzimidazoles or cyclic aminals, which are not suitable for our purpose. We thus chose *p*-DAPA as a starting point for synthetic variation while maintaining the single benzene core. By single-step reactions of the common *p*-DAPA intermediate, various carbonyl electrophiles were introduced to prepare different DAPA derivatives (**4–10**; Fig. 4). Compound **11** was prepared as a control to probe the involvement of ESIPT in the de-excitation pathway (vide infra). The chemical structures of the compounds **4–8** and **10** were unambiguously established by single-crystal X-ray diffraction analysis (Fig. 4a; Supplementary Figs. 3–8), which revealed intramolecular N–H⋯O (blue dotted lines) and C–H⋯O (green dotted lines) hydrogen bonds that restrict torsional freedom of the peripheral groups.

With fully characterized DAPA derivatives in hand, we proceeded to investigate their photophysical properties (Fig. 4, Table 1, and Supplementary Fig. 9). For the mono-functionalized DAPA derivatives **8–10**, both absorption ($\lambda_{max,abs}$ = 445–456 nm) and emission ($\lambda_{max,em}$ = 556–581 nm) are blue-shifted relative to the parent *p*-DAPA. With both amine sites substituted with carbonyl groups, more pronounced spectral blue-shifts were observed for the di-functionalized **4–7** in both absorption ($\lambda_{max,abs}$ = 370–435 nm) and emission ($\lambda_{max,em}$ = 471–543 nm). Apparently, carbonyl functionalization reduces the donor strength of the amine group of *p*-DAPA, thereby widening the HOMO–LUMO gap. Both the excitation and emission energies show linear correlations with the Hammett parameter (Supplementary Fig. 10). The fluorescence images and the chromaticity coordinates of the DAPA molecules (Fig. 4) further demonstrate synthetic tunability of the basic *p*-DAPA scaffold for systematic color engineering through last-stage single-step synthetic operations.

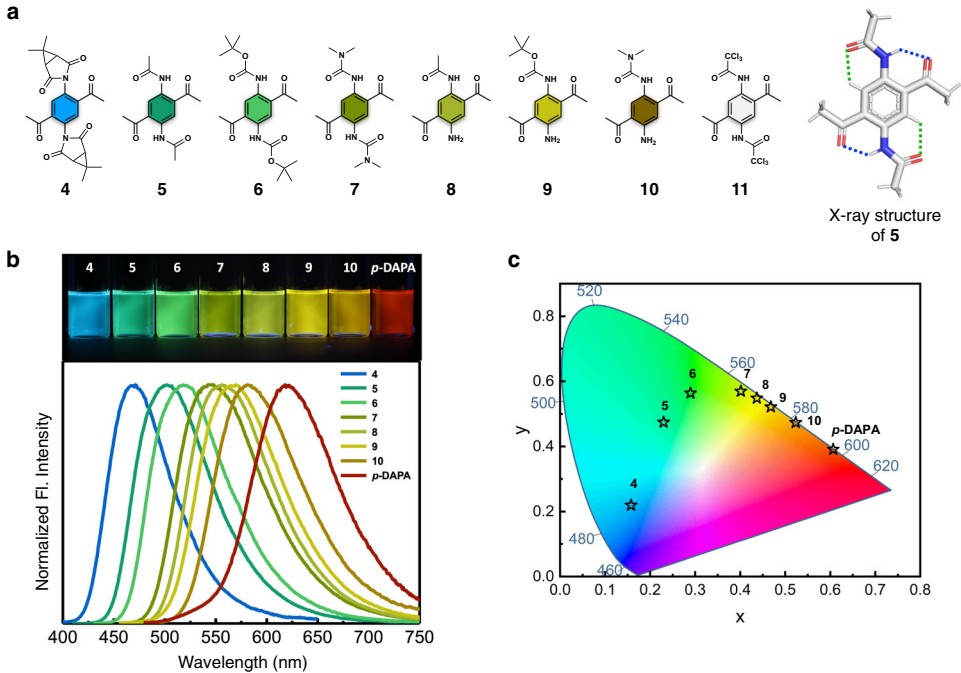

**Fig. 4 Full-color fluorophore library of DAPA. a** Chemical structures of *p*-DAPA derivatives **4–11**, along with capped-stick representation of **5** generated with crystallographically determined atomic coordinates. Hydrogen bonds are denoted by dotted lines. **b** Fluorescence images of **4–10** and *p*-DAPA in CHCl₃ under irradiation of 365 nm UV light (top), and normalized emission spectra (bottom). **c** Chromaticity coordinates (CIE) of **4–10** and *p*-DAPA in CHCl₃.

**Table 1 Photophysical properties of *o*-DAPA, *m*-DAPA, *p*-DAPA, and 4–10.**

|  | FW | $\lambda_{abs}{}^a$ (nm) | $\varepsilon$ (M$^{-1}$ cm$^{-1}$) | $\lambda_{em}$ (nm) | $\Delta v$ (cm$^{-1}$) | $\Phi_F{}^b$ | $\tau$ (ns) | $k_r$ (10$^7$ s$^{-1}$) | $k_{nr}$ (10$^7$ s$^{-1}$) |
|---|---|---|---|---|---|---|---|---|---|
| *o*-DAPA | 192.22 | 432 | 5160 | 531 | 4320 | 0.26 | 6.44 | 4.0 | 12 |
| *m*-DAPA | 192.22 | 350 | 7050 | - | - | - | - | - | - |
| *p*-DAPA | 192.22 | 482 | 2890 | 618 | 4570 | 0.06 | 1.84 | 3.3 | 51 |
| **4** | 436.46 | 370 | 3410 | 471 | 5800 | 0.18 | 5.83 | 3.1 | 14 |
| **5** | 276.29 | 405 | 5810 | 502 | 4770 | 0.73 | 8.93 | 8.1 | 3.1 |
| **6** | 392.45 | 416 | 6670 | 518 | 4730 | 0.61 | 9.09 | 6.7 | 4.3 |
| **7** | 334.38 | 435 | 7210 | 543 | 4570 | 0.29 | 6.14 | 4.7 | 12 |
| **8** | 234.26 | 445 | 3990 | 556 | 4490 | 0.30 | 5.93 | 5.0 | 12 |
| **9** | 292.34 | 446 | 3660 | 566 | 4750 | 0.27 | 4.89 | 5.5 | 15 |
| **10** | 263.30 | 456 | 5070 | 581 | 4720 | 0.23 | 2.96 | 7.7 | 26 |

*a*The longest absorption maximum wavelengths. *b*Absolute fluorescence quantum yields determined by a calibrated integrating sphere system. All measurements are made for CHCl₃ solution samples.

We note that these DAPA-based fluorophores also exhibit consistently large Stokes shifts ($\Delta v = 4490–5800$ cm$^{-1}$), which is highly unusual for small molecules. In addition to the spectral shifts, carbonyl substitution also contributes to significant enhancement in the fluorescence quantum yield ($\Phi_F$) (Table 1). Pairwise comparison of homologous sets of molecules (i.e., **5** with **8**; **6** with **9**; **7** with **10**) reveals an increase in $\Phi_F$ with an increasing number of carbonyl substitution. This consistent trend could be rationalized by the restriction of internal bond rotations by intramolecular hydrogen bonds, which suppresses non-radiative decay through thermal motion (Fig. 4 and Supplementary Figs. 4–8). This interpretation is further supported by relatively small non-radiative decay rate constants ($k_{nr}$) of **5–7** compared with **8–10** and the parent *p*-DAPA (Table 1).

**Excited-state energy landscape and de-excitation pathways.** To understand the structure-dependent emission properties (Fig. 3), the ground- and excited-states of DAPA isomers were investigated by ab initio calculations (Fig. 5).

For this purpose, we employed the mixed-reference spin-flip time-dependent density functional theory (MRSF-TDDFT; MRSF for brevity) method developed recently by us[38,39]. Unlike conventional DFT or linear-response TDDFT (LR-TDDFT), MRSF-TDDFT provides the exact dimensionality of conical intersections, describes strongly correlated ground and excited systems, and eliminates spin contamination inherent in other SF methods[40,41]. Using the computed structures (see Supplementary Tables 4–6 for energies; Supplementary Figs. 12–14 for geometries), minimum energy paths (Fig. 5) were constructed on the excited-state potential energy surfaces (PESs).

As shown in Fig. 5a, c, the two emissive isomers, *o*- and *p*-DAPA, undergo dipole-allowed vertical excitations (S₀ → S₁) at 3.73 eV (oscillator strength $f = 0.3133$) and 3.25 eV ($f = 0.2921$), respectively. Here, the S₁ state resulted from a single-electron HOMO → LUMO transition. Following vertical transition at the Franck–Condon (FC) geometry, the molecule undergoes a barrierless relaxation to the local minimum S₁,min, which is emissive. The predicted emission oscillator strengths ($f = 0.2467$

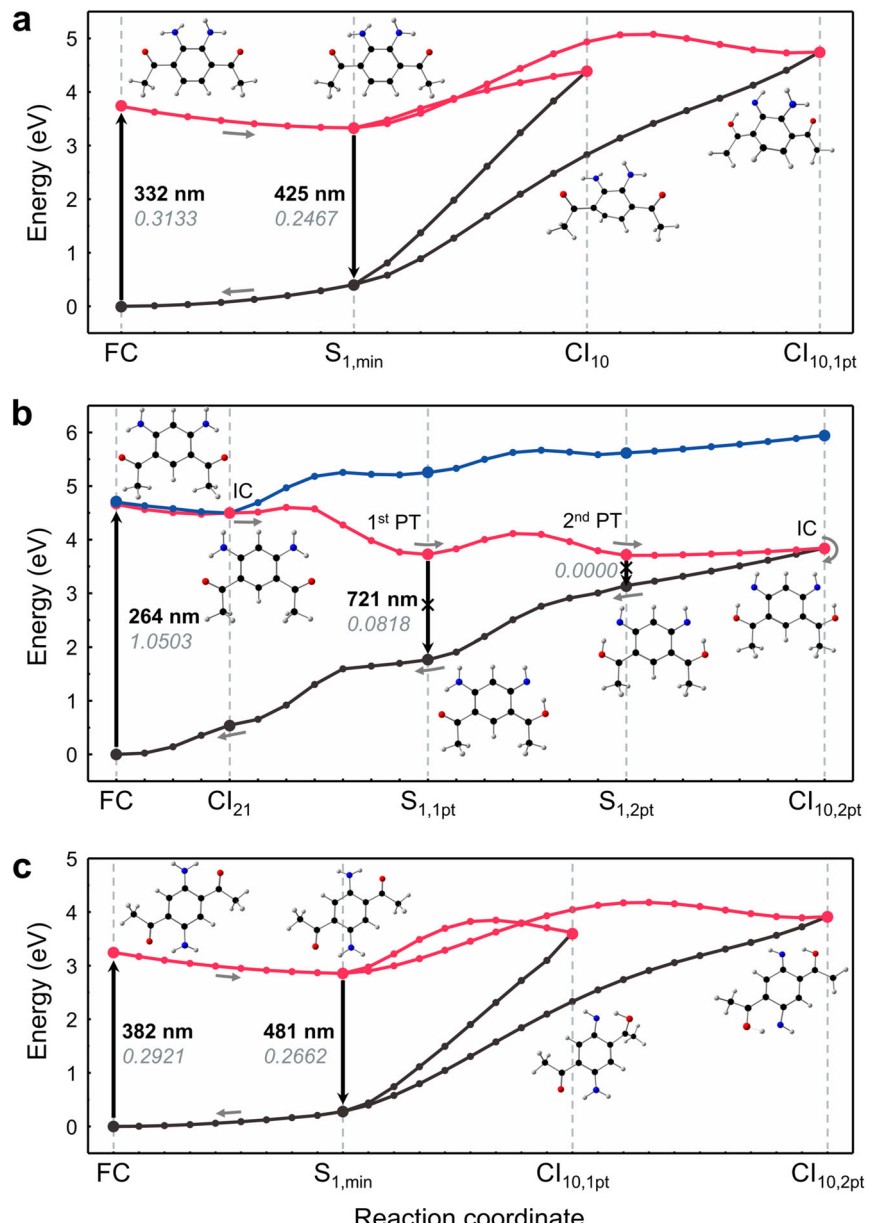

**Fig. 5 Potential energy surfaces.** Calculated $S_0$ (black), $S_1$ (red), and $S_2$ (blue) potential energy surfaces of $o$-DAPA (**a**), $m$-DAPA (**b**), and $p$-DAPA (**c**). The geometries were optimized at the MRSF/BH&HLYP/6-31 G* level of theory. Using computed structures, minimum energy paths (MEPs) were constructed, and optimized using the geodesic interpolation method[55]. For each transition, calculated wavelengths and oscillator strengths are shown in bold and italic, respectively. FC Franck–Condon region, CI conical intersection, IC internal conversion, PT proton transfer.

for $o$-DAPA; 0.2662 for $p$-DAPA) are only slightly lower than the absorption oscillator strengths.

On the $S_1$ PES of the $o$-DAPA (Fig. 5a), the two low-lying $S_1/S_0$ conical intersections, $CI_{10}$ and $CI_{10,1pt}$, are situated 0.66 eV and 1.01 eV above the FC point, and correspond to a puckering distortion of the benzene ring ($CI_{10}$) and single proton transfer ($CI_{10,1pt}$), respectively. The nonplanar geometry at $CI_{10}$ is best described as twist-boat, rather than genuine prefulvenic CI of benzene[37,42,43]. Since these two minimum energy conical intersections (MECIs) are poorly accessible, non-radiative decay is effectively suppressed. Consequently, $o$-DAPA stays at the $S_{1,min}$ geometry for an extended period of time, until it relaxes by fluorescence.

The emissive properties of $p$-DAPA can be explained in a similar fashion. As shown in Fig. 5c, the $S_{1,min}$ geometry is located 0.40 eV lower than the FC point. The two $S_1/S_0$ conical

intersections, corresponding to single ($CI_{10,1pt}$) and double ($CI_{10,2pt}$) proton transfer, are located 0.37 eV and 0.66 eV, respectively, above the FC point. This energy landscape effectively precludes a rapid non-radiative relaxation, and explains the fluorescence of $p$-DAPA from $S_{1,min}$.

A markedly different situation is encountered upon photo-excitation of $m$-DAPA (Fig. 5b). The vertical transition occurs to the $S_2$ state at 4.70 eV, which is almost 1 eV higher than the vertical excitation energy (VEE) of either $o$-DAPA or $p$-DAPA. Both the high VEE and the large oscillator strength ($f = 1.0503$) of $m$-DAPA are consistent with the experimentally observed short-wavelength absorption ($\lambda_{max,abs} = 276$ nm) with large absorptivity ($\varepsilon = 5.12 \times 10^4 \, M^{-1} \, cm^{-1}$) (Fig. 3).

As shown in Fig. 5b, slightly below the bright $S_2$ state lies a dark $S_1$ state (VEE = 4.66 eV). Due to this proximity, an $S_2 \rightarrow S_1$ internal conversion is possible, which is mediated by the conical

intersection CI$_{21}$ at 4.50 eV. In addition to this bright-to-dark internal conversion, geometric relaxation to optically dim local minima S$_{1,1pt}$ (4.07 eV) and S$_{1,2pt}$ (3.73 eV) can occur by a single proton transfer or a double proton transfer, respectively. Both S$_{1,1pt}$ and S$_{1,2pt}$ lie considerably lower in energy than the FC point, and have very low oscillator strengths of 0.0818 and 0.0000, respectively. In the proximity of S$_{1,2pt}$, conical intersection CI$_{10,2pt}$ (3.84 eV) is available, which corresponds to double proton transfer. This MECI can mediate non-radiative population transfer to the ground-state, thereby rendering *m*-DAPA non-fluorescent.

What could be the fundamental difference between the emissive *o*-DAPA and *p*-DAPA vs non-emissive *m*-DAPA? For *o*-DAPA and *p*-DAPA, the low-lying local minima in the S$_1$ state are easily accessible from the FC geometry, and effectively protected against non-radiative internal conversion, as the conical intersections are located considerably higher in energy. In contrast, *m*-DAPA is capable of rapid internal conversion to the dark excited-state or to the ground-state. The latter process is mediated by the low-lying conical intersections involving double proton transfer of the N–H protons. Translated into a more intuitive chemical model, unlike *o*-DAPA or *p*-DAPA, each amino group of *m*-DAPA is located at the *ortho*- and *para*-position of the electron-withdrawing acetyl groups. The increased acidity of the N–H bond enhances the mobility of the proton in the excited-states, such that ESIPT leads to the conical intersection.

**Excited-state antiaromaticity as the origin of the large Stokes shift.** A large Stokes shift of the molecular fluorescence is usually ascribed to ICT[5,6] or ESIPT[17–19]. In line with this prevailing paradigm, the unusual long-wavelength emission of SBF is often interpreted as de-excitation from the ICT states originating from HOMO–LUMO asymmetry[29–32]. Considering the small size of the benzene core that does not allow for proper charge separation, the validity of such an explanation is questionable. In fact, our theoretical study (Fig. 5) predicts that the large Stokes shift of the parent *p*-DAPA arises from geometric relaxation in the S$_1$ state, which involves neither ICT nor ESIPT. This interpretation is also supported by the fact that the Mulliken charge distribution of S$_{1,min}$ does not differ much from that of FC (Supplementary Fig. 24).

We propose that the unusual photophysics of DAPA fluorophores originates from the excited-state antiaromaticity (ESAA) of the benzene core itself. According to Baird's rule, the lowest triplet (T$_1$) and singlet (S$_1$) excited-states of small annulenes have antiaromatic characteristics[36,37,44–46]. For example, benzene becomes antiaromatic in the singlet excited-state[36,37], which results in a strong destabilization of the S$_1$ state at the FC geometry. This destabilization can be relieved through a substantial bond length redistribution leading to S$_1$ local minimum. This process, in turn, results in a large Stokes shift when benzene reverts to the ground-state. In support of this notion, the parent benzene ring system has a large Stokes shift of 4410 cm$^{-1}$ even without any substituents. With increasing ring fusion and expansion of the π-conjugation, a systematic decrease in the Stokes shift is observed (Supplementary Fig. 25).

To examine the validity and general applicability of this intuitive model, the energy and the nucleus-independent chemical shift (NICS)[47] values were calculated for the ground (S$_0$) and excited (S$_1$) states of the DAPA-based SBFs. The S$_0$ and S$_1$ geometries were optimized at the MRSF/BH&HLYP/6-31 G* level, and the results of the NICS calculations are summarized in Fig. 6 and Table 2. The geometric aspects of ground-state aromaticity (GSA) were also analyzed by comparing the

harmonic oscillator model of aromaticity (HOMA)[48] indices at S$_{0,min}$ and S$_0$@S$_{1,min}$.

For *p*-DAPA, the S$_0$ ground-state is aromatic (NICS(1)$_{zz}$ = −24.0 ppm) but becomes antiaromatic (NICS(1)$_{zz}$ = +31.2 ppm) upon excitation to S$_1$ FC. This ESAA becomes relieved as S$_{1,min}$ (NICS(1)$_{zz}$ = +8.8 ppm) is reached by bond length redistribution. The radiative decay takes place at S$_{1,min}$. The large difference between the vertical excitation energy at the FC point (3.25 eV) and the vertical de-excitation energy at the S$_{1,min}$ geometry (2.58 eV) accounts for the large Stokes shift.

The graphical representations of the calculated NICS(1)$_{zz}$ grids (Fig. 6b) also indicate that *p*-DAPA is aromatic in the ground electronic state, but becomes noticeably antiaromatic in the excited-state at the FC geometry. Upon geometric relaxation to the S$_1$ local minimum, the excited-state energy is lowered (Fig. 6a, top). On the other hand, the energy of the ground-state is raised at S$_0$@S$_{1,min}$, with attenuated GSA relative to S$_{0,min}$ (Fig. 6a, bottom). A decrease in the HOMA value (Fig. 6c, d) as a geometric indicator of aromaticity supports this notion.

Taken together, relaxation of ESAA and loss of GSA collectively lead to an overall narrowing of the S$_1$–S$_0$ energy gap of the radiative decay, as manifested by the large Stokes shift observed experimentally. With consistent trends in the changes of NICS(1)$_{zz}$ (Table 2) and HOMA (Supplementary Table 10) values, *o*-DAPA and **5–10** also operate by similar de-excitation mechanism.

Peripheral functional groups mediate proton transfer (PT)[49,50] or proton-coupled electron transfer (PCET)[51] to relieve the antiaromaticity of π-systems in the excited-state. We now show that hydrogen bonds can also assist in the relief of ESAA by bond length redistribution without invoking extensive structural rearrangement. For emissive DAPA derivatives, a significant shortening of the $d_{N–H···O}$ interatomic distance is observed consistently (Fig. 6c, d, and Supplementary Fig. 26), along with an increase in the electron density at the bond critical point (Supplementary Table 11). Such strengthening of hydrogen bonds could provide an extra stabilization to alleviate ESAA without significant charge redistribution (ICT) or proton transfer (ESIPT).

To check the validity of our theoretical model, a number of experimental studies were carried out. Unlike **4–10**, the control molecule **11** (Fig. 4a) displays dual emission (Supplementary Fig. 30). With the strongly electron-withdrawing Cl$_3$C(=O)–substituent, the N–H proton of **11** becomes sufficiently acidic to open up a pathway to intramolecular proton transfer, as manifested by the broad and longer-wavelength ESIPT emission ($\lambda_{max,em}$ = 579 nm, $\Delta\nu$ = 8290 cm$^{-1}$) along with the relatively sharp local emission ($\lambda_{max,em}$ = 482 nm, $\Delta\nu$ = 5030 cm$^{-1}$). In stark contrast, only local emission was observed for **4–10** as a single sharp band which mirrors the absorption (Supplementary Fig. 9), thus ruling out the involvement of ESIPT pathway.

To probe the ICT characteristics, the dependence of the emission energy on the solvent polarity was also investigated. As summarized in Supplementary Table 12, the DAPA fluorophores show no pronounced solvatochromism. Except for DMSO or EtOH, which disrupt hydrogen bonds, the Stokes shifts of the representative DAPA fluorophores (*o*-DAPA, *p*-DAPA, and **5**) are essentially invariant to changes in the solvent polarity, and remain at around 4500 cm$^{-1}$. Based on the Lippert–Mataga equation (Supplementary Fig. 31)[52], the excited-state dipole moments of *p*-DAPA and **5** are estimated to be 5.1 D and 6.0 D, respectively, which are substantially lower than those of typical ICT fluorophores (~ 20 D)[5,16]. Structurally related SBFs[29–32] also exhibit the characteristics of solvent-independent emission and small dipole moment in the excited-state, which are distinct from typical ICT fluorescence[5,16]. To

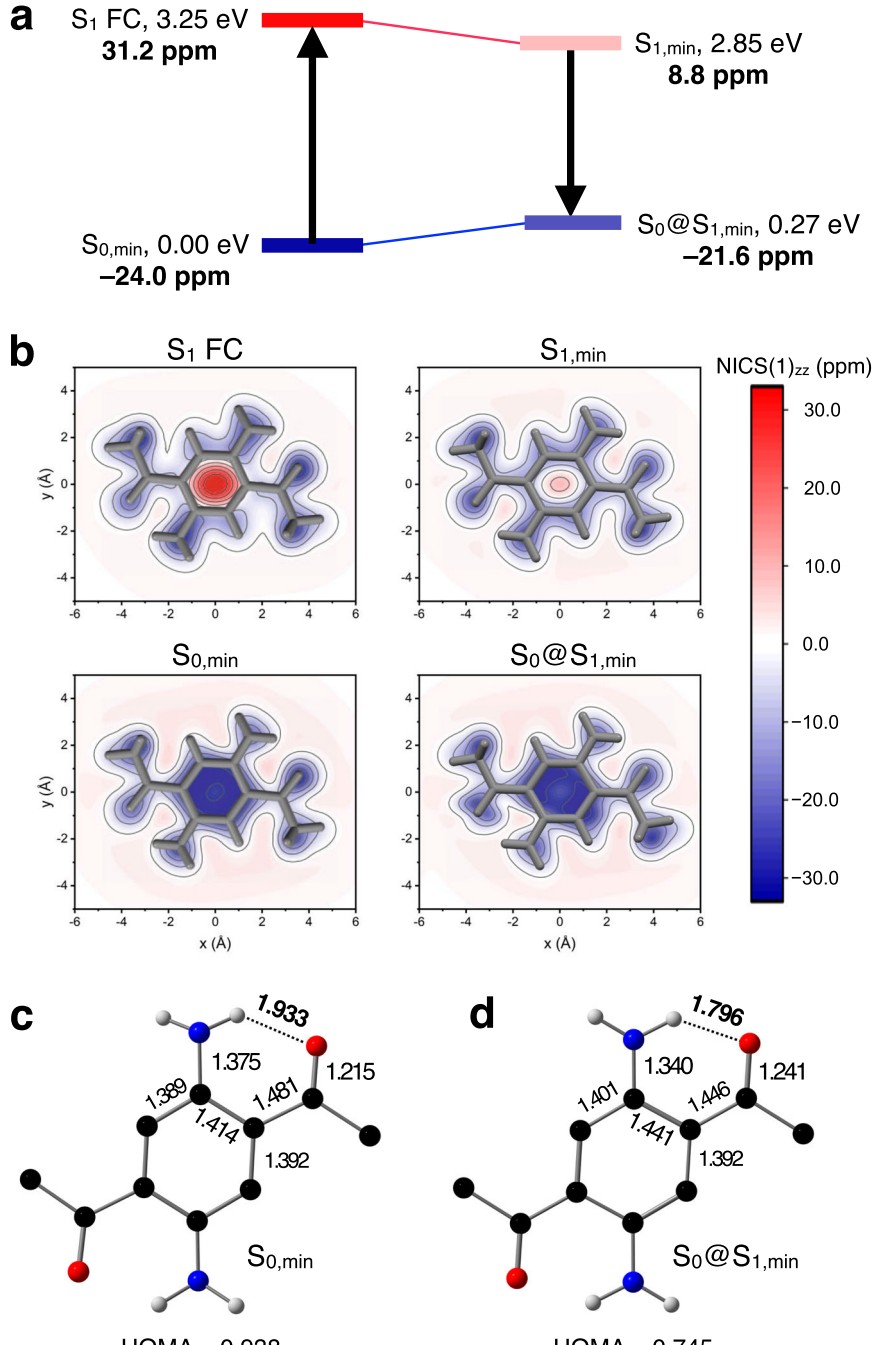

**Fig. 6 Relief of excited-state antiaromaticity assisted by intramolecular hydrogen bonds. a** Schematic energy diagram with calculated NICS(1)$_{zz}$ values (in bold) at the optimized geometries of *p*-DAPA. **b** Calculated NICS(1)$_{zz}$ grids parallel to the molecular plane of *p*-DAPA. **c, d** Bond lengths (Å) and HOMA value of *p*-DAPA at the S$_{0,min}$ (**c**), and S$_0$@S$_{1,min}$ (**d**) geometry. FC Franck–Condon, min minimum, NICS nucleus-independent chemical shift, HOMA harmonic oscillator model of aromaticity.

demonstrate the general applicability of our mechanistic model, the same computational protocols were employed to probe the ground- and excited-states of amino–sulfonyl[30], and amino–ester[31] SBFs (Supplementary Fig. 32). Both SBFs become antiaromatic upon vertical excitation, and the antiaromaticity is relieved with subsequent geometric relaxation to S$_{1,min}$. We thus conclude that the large Stokes shifts of SBFs originate from the relief of antiaromaticity of benzene core itself, without involvement of significant charge redistribution or nucleus movement in the peripheral groups.

## Discussion

By a combination of experimental and theoretical studies, we show that a single benzene core is sufficient to achieve long-wavelength fluorescence, without the need for large and extended π-conjugation. Our in-depth quantum chemical calculations of the three different regioisomers of DAPA also revealed a critical functional role played by the relative positioning of HBD–HBA pairs that dictates the branching between emissive vs non-emissive pathways in the excited-states, which is unveiled for the first time in the SBF systems.

**Table 2 Calculated NICS(1)$_{zz}$ values for emissive DAPA isomers and derivatives[a].**

|  | $S_{0,min}$ | $S_1$ FC | $S_{1,min}$ | $S_0@S_{1,min}$ |
|---|---|---|---|---|
| *o*-DAPA | −23.0 | 26.6 | 9.0 | −22.4 |
| *p*-DAPA | −24.0 | 31.2 | 8.8 | −21.6 |
| **5** | −25.4 | 77.7 | 33.0 | −23.2 |
| **6** | −27.1 | 102.0 | 44.6 | −24.2 |
| **7** | −24.8 | 59.4 | 25.1 | −22.9 |
| **8** | −24.1 | 17.8 | 6.0 | −21.6 |
| **9** | −24.0 | 19.6 | 7.1 | −21.4 |
| **10** | −24.4 | 15.3 | 4.9 | −21.7 |

[a]The NICS values were calculated using $S_0$ and $S_1$ wavefunctions obtained at the CASSCF(2,2)/6-31 G* level of theory.

Through facile synthetic modification of the peripheral HBD groups, the HOMO–LUMO energy gap of DAPA derivatives can be varied systematically. With structure-independent large Stokes shifts, these new fluorophores cover the entire visible color spectrum. We found that their unusually large Stokes shifts (ca 4500 cm$^{-1}$) originate from significant relief of the excited-state antiaromaticity by bond length redistribution. In hindsight, it makes perfect sense that the effects of antiaromaticity should be the most pronounced for the smallest benzene ring. With ring fusion and substitution, however, the antiaromaticity becomes less apparent for typical fluorophores having large and extended π-conjugation. Therefore, they inevitably rely on ICT or ESIPT to realize large Stokes shifts. Our work on the lightest red-emitter paves the way for the applications of these new SBFs for bioimaging and light-emitting devices, which are currently underway in our laboratory.

## Methods

Synthetic procedures and characterization of DAPA fluorophores reported in this work are provided in Supplementary Information.

**Physical measurements.** $^1$H NMR and $^{13}$C NMR spectra were recorded on a 500 MHz Varian/Oxford As-500 spectrometer. Chemical shifts were referenced to internal standard of tetramethylsilane (as δ = 0.00 ppm). High-resolution electrospray ionization (ESI) mass spectra were obtained on a Thermo Scientific LTQ Orbitrap XL mass spectrometer. FT-IR spectra were recorded on a Shimadzu IRTracer-100 FT-IR Spectrophotometer. Electronic absorption spectra were recorded on an Agilent 8453 UV–vis spectrophotometer with ChemStation software. Fluorescence spectra were recorded on a Photon Technology International Quanta-Master 400 spectrofluorometer with FelixGX software. The quantum yields were determined by using an integrating sphere attached to the instrument. Time-resolved photoluminescence (PL) decay measurements were made on a Edinburgh FLS-920 equipped with a 450 nm diode laser (EPL-450).

**Computational studies.** All MRSF/BH&HLYP/6-31G* calculations were performed using the local GAMESS[53] package. Solvent effects were included using the polarizable continuum model (PCM). The minimum energy conical intersections were optimized by a branching plane updating algorithm[54]. MRSF-TDDFT is capable of producing the correct double-cone topology of the intersections and describing the geometry of the lowest-energy conical intersections and their relative energies with accuracy matching that of the best multireference wavefunction ab initio methods[39]. BH&HLYP functional was employed to provide the best performance of the MRSF methodology as verified by previous benchmarking studies[40,41]. The potential energy surfaces (PESs) were constructed by the recently reported geodesic interpolation method[55], which reformulates the problem of existing interpolation methods by searching the geodesic curve on the Riemannian manifold. The NICS(1)$_{zz}$ values were computed using complete-active-space self-consistent field calculation with gauge-including atomic orbitals (CASSCF-GIAO) methodology[56] within the 6-31 G* basis set implemented in the Dalton package[57,58]. The $S_0$ and $S_1$ wavefunctions obtained from CASSCF/6-31 G* were used to calculate the NICS(1)$_{zz}$ values at MRSF/BH&HLYP/6-31 G* optimized geometry. The small active space seems to be sufficient here, since the $S_1$ states of the DAPA derivatives are mainly accessed by the one-electron HOMO–LUMO transitions[59].

## Data availability

The X-ray crystallographic coordinates for structures reported in this study have been deposited at the Cambridge Crystallographic Data Centre (CCDC), under deposition numbers 2054917 (*o*-DAPA), 2054916 (*m*-DAPA), 2054922 (*p*-DAPA), 2054920 (**4**), 2054914 (**5**), 2054915 (**6**), 2054921 (**7**), 2054918 (**8**), and 2054919 (**10**). These data can be obtained free of charge from The Cambridge Crystallographic Data Centre via www.ccdc.cam.ac.uk/data_request/cif. Experimental procedures, NMR spectra, supplementary figures, and supplementary tables are available in the Supplementary Information.

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

## Acknowledgements

This work was supported by the Samsung Science and Technology Foundation (SSTF-BA1701-09 to D.L.), and the National Research Foundation of Korea (NRF; 2019H1D3A2A02102948 to M.F.; 2020R1A5A1019141 to C.H.C.). H.K. is a recipient of the NRF Global Ph.D. Fellowship (2018H1A2A1060348).

## Author contributions

H.K. and W.P. contributed equally to this work. H.K. and D.L. initialized the project. H.K. and Y.K. synthesized and characterized the compounds. D.L. supervised the experimental component. W.P. under the guidance of M.F. and C.H.C. performed MRSF-TDDFT and CASSCF calculation. All authors contributed in drafting the manuscript.

## Competing interests

The authors declare no competing interests.
