## [Peer Review File · Nature Communications]

Relief of Excited-State Antiaromaticity Enables the Smallest Red EmitterREVIEWERS' COMMENTS

Reviewer #1 (Remarks to the Author):

The manuscript by Kim et al is one of the most interesting I have read so far this year. Building on a serendipitous synthesis of DAPA and observation of its emissive property they undertake a careful investigation using both experimental and computational approaches to explore the fundamental properties of these chromophores and further design a DAPA fluorophore library. Yet, in contrast to the authors I think the emission process has similarities with ESIPT as I suspect that the emissions in o- and p-DAPA are due to what I would describe as "halted ESIPT" (see below). This in itself is very interesting. I think the findings reported will be highly appreciated by researchers in several areas ranging from theoretical chemists interested in excited state aromaticity effects to biophysical chemists interested in new fluorophores for bioimaging applications. Still, I have some items that should be addressed by the authors before I can recommend acceptance.

1: In the Introduction (p. 2) the authors have not addressed that gain of excited state aromaticity in some benzannelated 8π -electron species (e.g., dibenzoxepins) is a sixth option to achieve a system that exhibits a large Stokes shift. This was first observed by Wan and Shukla in 1993 (JACS, 1993, 115, 2990). The ESA gain was confirmed more recently through computations by Toldo et al (ChemPlusChem 2019, 84, 712) and the compounds were further explored by Saito and co-workers (JACS, 2020, 142, 14985). This should be mentioned on page 2 as the sixth strategy to design systems with large Stokes shifts.

2: The ESAA relief of the benzene core is crucial for their findings, as noted at the end of the Introduction. In this context I propose to refer to the recent study by Slanina et al where ESAA relief in the singlet excited benzene itself was investigated (JACS 2020, 142, 10942). It can be noted that the ESAA character in benzene as determined by NICS (99.5 ppm vertically and 80.5 ppm relaxed S1 geometry) is significantly larger than in most of the DAPA molecules (the exceptions being compounds 5 - 7, see Table 2). As also the vertically excited structures are less S1 antiaromatic than benzene there seems to be also a purely electronic relaxation of the ESAA character via the substituents of the DAPAs. The puzzling cases are compounds 5 - 7; why are those more strongly S1 state antiaromatic than the others?

3: Looking at the S0 and S1 state geometries shown in Figure 6c-d I would describe the structural change which they observed in S1 as a "halted ESIPT" or "partial ESIPT", yet here I would like to know the symmetries of p-DAPA in S0 and S1 according to the computations. Are the geometries C2 or even C2h symmetric?

Still, with two pairs of neighboring amino and carbonyl groups (formally allowing for ESIPT) there are two moieties that will assist in the ESAA relief. As a consequence, the S1 state molecule will not need to undergo a similarly extensive structural rearrangement as it would when having merely one pair of adjacent amino and carbonyl groups where the ESIPT reaction occurs. In this context they should compare to two recent studies connecting ESIPT reactions with ESAA relief; see PNAS, 2019, 116, 20303 and PCCP, 2019, 21, 11608.

4: As the authors predict that o- and p-DAPAs could be used in bioimaging applications I would like to know about their spectroscopic features when in water solutions. Do they still emit efficiently or is the emission quenched by intermolecular bonding between the amino and carbonyl groups and the surrounding water?

5: Then, why did they select the BH&HLYP functional? This is not explained in neither the manuscript or the SI.

6: I would like to see NICS values calculated with a larger basis set, including diffuse functions.

7: With regard to the conical intersections shown in Figure 5 and discussed on page 9, do they correspond to the prefulvenic conical intersection observed for benzene?

Minor language items:

8: In the Abstract on page 1 should be "excited-state antiaromaticity" instead of "excited-state aromaticity"

9: In my view the subtitle of the section starting at the end of page 10 should be "Excited-State Antiaromaticity as the Origin ..." instead of "Excited State Aromaticity as the Origin ...".

10: At the end of the first paragraph on page 14 it says "Strokes" instead of "Stokes".

Reviewer #2 (Remarks to the Author):

The manuscript entitled "Relief of Excited State Antiaromaticity Enables the Smallest Fluorophore" is well-written and deals with the highly relevant question of how to make smaller fluorophores with red(-shifted) emission. I find the title a bit confusing and one word short. If it concerns "the Smallest Red Fluorophore" it would be a lot clearer to me.

Small red-shifted fluorophores, single benzene fluorophores (SBFs) are known already. The authors present a new class of SBFs, worked out the organic synthesis skilfully and synthesised a set of related compounds, whose emission cover the spectrum from 470 to 620 nm. The best examples presented by the authors are not extraordinary as such; the claim of having the fluorophore with the lowest molecular weight is valid (192 vs 258 from Ref 28), that of the longest emission wavelength as well, but not by a wide margin. Also the fluorescence quantum yields, 8% for p-DAPA are not exceptional.

The true value of this work is in constructing a proper explanation of the red shift of the absorption and more importantly, the exceptionally large Stokes shift that results in an even further red-shifted emission.

Since computational chemistry is not my discipline, it is hard for me to judge the quality of the calculations. Nevertheless the explanation of the results from these calculations are clear to me, the line of reasoning is correct, and above all, the refutation of alternative explanations (for the large Stokes shifts) are convincing. The concept of antiaromaticity in the excited state (as a means to decrease the energy content of the excited state by relaxation) is new to me and should be particularly strong for small aromatic molecules. Also experimental evidence is provided with a solvatochromic investigations on the DAPA compounds (Table 12) and a demonstration of the effect of ESIPT on compound 11.

In my opinion, this manuscript is original, complete and provides new insights for developing fluorescent organic molecules. I have only minor suggestions for textual improvement listed below. I recommend publication with minor revision.

Textual remarks:

Page 9, line 10:

Here, the S1 state corresponds to a single-electron HOMO → LUMO transition.

Wouldn't it be more correct to state that:

Here, the S1 state resulted from a single-electron HOMO → LUMO transition.

Page 10, line 9:

Fluorescence spectra were recorded on a Photon Technology International Quanta-Master 400 spectrouorometer with FelixGX software.

Change spectrouorometer to either spectrofluorometer or spectrophotometer

In Supplementary Table 3, with entry 1, reference 28 should be mentioned (this one is from the German edition)

Reviewer #3 (Remarks to the Author):

This work deals with the interesting and practically very important question of how to create large Stokes shifts and efficient fluorescent emitters from small aromatic hydrocarbons. The authors achieve this remarkable goal by means of diacetylphenylenediamine (DAPA) as basic compound obviating the use of larger polycyclic aromatic hydrocarbons. This is certainly a major progress, which certainly will influence the development of optical biosensors. The work describes the experimental characterization of the compounds and also provides a theoretical explanation of the reason for the large Stokes shifts by means of high-level quantum chemical calculations. In these calculations, minimum energy paths were constructed from the Franck-Condon region to minimum energy conical intersections (MECI) using an interesting mixed-reference spin-flip approach. The fluorescent o- and p-DAPA compounds show an uphill path to the intersections, whereas for the m-DAPA structure the MECI is energetically accessible via two proton transfer steps. Relief of antiaromaticity of the excited state as demonstrated via NICS calculations is invoked to rationalize the large Stokes shift. In summary, this is an interesting paper, which is clearly written and explained. I recommend publication of this work as it is.

Reviewer #1. We thank the reviewer for careful reading and positive evaluation of our manuscript. He/she commented that "*The manuscript by Kim et al is one of the most interesting I have read so far this year . . . I think the findings reported will be highly appreciated by researchers in several areas ranging from theoretical chemists interested in excited state aromaticity effects to biophysical chemists interested in new fluorophores for bioimaging applications.*" Listed below are our point-by-point responses to her/his helpful comments and insightful suggestions.

1. In the Introduction (p. 2) the authors have not addressed that gain of excited state aromaticity in some benzannelated 8π -electron species (e.g., dibenzoxepins) is a sixth option to achieve a system that exhibits a large Stokes shift. This was first observed by Wan and Shukla in 1993 (JACS, 1993, 115, 2990). The ESA gain was confirmed more recently through computations by Toldo et al (ChemPlusChem 2019, 84, 712) and the compounds were further explored by Saito and co-workers (JACS, 2020, 142, 14985). This should be mentioned on page 2 as the sixth strategy to design systems with large Stokes shifts.

- We appreciate the reviewer's helpful comments. The suggested references have been included as #26–28, and the main text revised accordingly:

"Existing strategies to realize large Stokes shift rely on (i) intramolecular charge transfer (ICT)^{5,6,16}, (ii) excited-state intramolecular proton transfer (ESIPT)^{17–19}, (iii) fluorescence resonance energy transfer (FRET)^{20,21}, (iv) desymmetrization of the molecular structure^{6,22–24}, (v) excimer/exciplex emission²⁵, or (vi) excited-state planarization of the benzannelated 8π system^{26–28}."

(Page 2, Paragraph 2)

26. Shukla, D. & Wan, P. Evidence for a planar cyclically conjugated 8π system in the excited state: Large Stokes shift observed for dibenz[b,f]oxepin fluorescence. *J. Am. Chem. Soc.* **115**, 2990–2991 (1993).

27. Toldo, J., El Bakouri, O., Solá, M., Norrby, P.-O. & Ottosson, H. Is excited-state aromaticity a driving force for planarization of dibenzannelated 8π -electron heterocycles? *ChemPlusChem* **84**, 712–721 (2019).

28. Kotani, R. et al. Controlling the S_1 energy profile by tuning excited-state aromaticity. *J. Am. Chem. Soc.* **142**, 14985–14992 (2020).

2-1. The ESAA relief of the benzene core is crucial for their findings, as noted at the end of the Introduction. In this context I propose to refer to the recent study by Slanina et al where ESAA relief in the singlet excited benzene itself was investigated (JACS 2020, 142, 10942).

- ▶ As suggested by the reviewer, the article by Slanina et al. is cited as reference #37 in the revised manuscript.

"We found that the excited-state antiaromaticity of the benzene core itself^{36,37}, rather than the typically assumed ICT or ESIPT, is responsible for their peculiar photophysical properties."

(Page 3, Paragraph 1)

37. Slanina, T. et al. Impact of excited-state antiaromaticity relief in a fundamental benzene photoreaction leading to substituted bicyclo[3.1.0]hexenes. *J. Am. Chem. Soc.* **142**, 10942–10954 (2020).

2-2. It can be noted that the ESAA character in benzene as determined by NICS (99.5 ppm vertically and 80.5 ppm relaxed S₁ geometry) is significantly larger than in most of the DAPA molecules (the exceptions being compounds 5 - 7, see Table 2). As also the vertically excited structures are less S₁ antiaromatic than benzene there seems to be also a purely electronic relaxation of the ESAA character via the substituents of the DAPAs. The puzzling cases are compounds 5 – 7; why are those more strongly S₁ state antiaromatic than the others?

- ▶ As the reviewer noted, the NICS(1)_{zz} values of **5–7** at S₁ FC and S_{1,min} indeed seem to be unusually large compared with those of other DAPA molecules. In the special case of **6**, the vertical value of 102.0 ppm is even larger than that of benzene (99.5 ppm). However, unlike benzene (80.5 ppm), the value at the relaxed S₁ geometry (S_{1,min}) of **6** is dramatically reduced to 44.6 ppm, indicating a strong impact of the substituents. While changes in the relative NICS values faithfully represent the essential feature of ESAA, its absolute magnitude at FC might still be a concern.
- ▶ To rule out computational artifacts, we compared the entire NICS(1)_{zz} grids, rather than a single NICS(1)_{zz} value, of **5** at S₁ FC with those of *p*-DAPA and **8** (please see **Fig. 6**, and **Supplementary Figs. 28** and **29** of the revised SI for these and additional data). As shown in **Response Fig. 1** on the next page, the NICS(1)_{zz} grids of **5** are centrosymmetric and continuous at S₁ FC, similar to those of *p*-DAPA and **8**.

Response Fig. 1 | NICS(1)_{zz} grids of (a) *p*-DAPA, (b) **5**, and (c) **8** at S₁ FC. Note that the color scale is different for **5**.

- At the same time, the numerical value of NICS_{zz} depends strongly on the measuring point, i.e. the elevation above the ring plane. We thus tracked the changes of the NICS_{zz} values as a function of the elevation above the center of the benzene ring. As shown in **Response Fig. 2** below, the NICS_{zz} values decrease smoothly as the elevation increases from 0 Å to 1.5 Å. Such a behavior is typical for an antiaromatic annulene¹. Based on these calculations, we can confidently rule out the involvement of computational artifacts.

Response Fig. 2 | Plots of NICS_{zz} values of *o*-DAPA, *p*-DAPA, **5**, and **8** at S₁ FC as a function of the distance above the center of benzene ring.

- It should be noted that the NICS index is just one of the many criteria to evaluate (anti)aromaticity. While it is a convenient and widely adopted indicator of aromatic stability, the correlation is not always guaranteed². It is also known that the NICS values tend to be overestimated for antiaromatic molecules in the excited states³. Therefore, we believe that the NICS parameter can be best used to describe the aromatic vs. antiaromatic properties of the same molecule in a relative sense. A direct comparison of its absolute numerical values per se between different molecules might be less meaningful. In terms of the general trend, the NICS values become attenuated upon geometric relaxation from S₁ FC to S_{1,min}. This behavior is observed consistently and reproducibly across all DAPA molecules (see **Table 2**), which further supports the validity and general applicability of our ESAA relaxation model shown in **Fig. 6**.
- To augment the inherent limitations of NICS method, we also employed HOMA (harmonic oscillator model of aromaticity) index as a geometric criterion of aromaticity. For all of the DAPA series molecules, we observed a significant and consistent decrease in the HOMA index at S₀@S_{1,min} compared with that at S_{0,min}, which unambiguously points toward the

¹ Stanger, A. Nucleus-independent chemical shifts (NICS):Distance dependence and revised criteria for aromaticity and antiaromaticity. *J. Org. Chem.* **71**, 883–893 (2006).

² Chen, Z., Wannere, C. S., Corminboeuf, C., Puchta, R. & Schleyer, P. v. R. Nucleus-independent chemical shifts (NICS) as an aromaticity criterion. *Chem. Rev.* **105**, 3842–3888 (2005).

³ Karas, L. J., Wu, C.-H., Ottosson, H. & Wu, J. I. Electron-driven proton transfer relieves excited-state antiaromaticity in photoexcited DNA base pairs. *Chem. Sci.* **11**, 10071–10077 (2020).

attenuation of ground-state aromaticity (**Fig. 6** and **Supplementary Table 10**). It should be noted that such feature does not stand out by comparing the NICS values alone, since the change in the magnitude of NICS values at S_0 are less remarkable (**Table 2**).

- ▶ A consistent mechanistic picture thus emerges from a combination of electronic (i.e. NICS) and geometric (i.e. HOMA) analysis: relaxation of the ESAA and attenuation of the GSA work in the same direction to produce unusually large Stokes shifts of the DAPA molecules.

3-1. Looking at the S_0 and S_1 state geometries shown in Figure 6c-d I would describe the structural change which they observed in S_1 as a "halted ESIPT" or "partial ESIPT", yet here I would like to know the symmetries of *p*-DAPA in S_0 and S_1 according to the computations. Are the geometries C_2 or even C_{2h} symmetric?

- ▶ Without any geometric constraints, the optimized geometries of *p*-DAPA and **5–7** in S_1 (= $S_{1,\min}$) all converge to C_{2h} symmetry (**Figs. 6c and 6d**, and **Supplementary Figs. 14–15**). As such, strengthening of the intramolecular hydrogen bonds occurs simultaneously for the two N–H \cdots O pairs. Apparently, the two pairs of hydrogen bonding donor–acceptor (HBD–HBA) work in concert to avoid (or "halt") ESIPT. Please see below for a continued discussion on this topic.

3-2. Still, with two pairs of neighboring amino and carbonyl groups (formally allowing for ESIPT) there are two moieties that will assist in the ESAA relief. As a consequence, the S_1 state molecule will not need to undergo a similarly extensive structural rearrangement as it would when having merely one pair of adjacent amino and carbonyl groups where the ESIPT reaction occurs.

- ▶ This is an interesting idea. As the reviewer suggested, the two pairs of HBD–HBA within DAPA might provide sufficient geometric distortion to relieve ESAA without invoking an extensive structural rearrangement such as complete proton transfer. At the same time, the fate of the excited state also depends on how these two pairs of HBD–HBA are positioned around the central benzene ring. Right after FC, *m*-DAPA crosses a unique conical intersection (CI_{21}) which takes it into a dark state leading to multiple ESIPT steps and eventual non-radiative decay (**Fig. 5b**). In contrast, its isomeric form *o*-DAPA or *p*-DAPA does not fall into such ESIPT pathways, and becomes emissive. We tentatively conclude that both the number and relative positioning of the HBD–HBA pairs dictate the initial ultrafast non-adiabatic electronic transition as the entry point to ESIPT, which is allowed for *m*-DAPA, but seems to be "halted" for *o*-DAPA and *p*-DAPA. A detailed understanding of the mechanistic branching between these competing reaction pathways would require extensive NAMD (non-adiabatic molecular dynamics) simulation, which is a topic of our future studies. We thank the reviewer for the insightful comments.

3-3. In this context they should compare to two recent studies connecting ESIPT reactions with ESAA relief; see PNAS, 2019, 116, 20303 and PCCP, 2019, 21, 11608.

- ▶ We thank the reviewer for bringing these relevant works to our attention, which are now included as references #47 and #48. With two references on relieving ESAA via CT (#49) or PCET (#50) also added, the revised text reads:

"Peripheral functional groups mediate proton transfer (PT)^{47,48}, charge transfer (CT)⁴⁹, or proton-coupled electron transfer (PCET)⁵⁰ to relieve the antiaromaticity or reinforce the aromaticity of π -systems in the excited-state. We now show that hydrogen bonds can also assist in the relief of ESAA by bond length redistribution without invoking extensive structural rearrangement . . ."

(Page 13, Paragraph 3)

47. Lampkin, B. J., Nguyen, Y. H., Karadakov, P. B. & VanVeller, B. Demonstration of Baird's rule complementarity in the singlet state with implications for excited-state intramolecular proton transfer. *Phys. Chem. Chem. Phys.* **21**, 11608–11614 (2019).

48. Wu, C. H., Karas, L. J., Ottosson, H. & Wu, J. I. Excited-state proton transfer relieves antiaromaticity in molecules. *Proc. Natl. Acad. Sci. U.S.A.* **116**, 20303–20308 (2019).

49. Kim, J. et al. Two-electron transfer stabilized by excited-state aromatization. *Nat. Commun.* **10**, 4983 (2019).

50. Karas, L. J., Wu, C.-H., Ottosson, H. & Wu, J. I. Electron-driven proton transfer relieves excited-state antiaromaticity in photoexcited DNA base pairs. *Chem. Sci.* **11**, 10071–10077 (2020).

4. As the authors predict that *o*- and *p*-DAPAs could be used in bioimaging applications I would like to know about their spectroscopic features when in water solutions. Do they still emit efficiently or is the emission quenched by intermolecular bonding between the amino and carbonyl groups and the surrounding water?

- ▶ As the reviewer inquired, fluorescence quantum yield in the polar environment is an important practical consideration for bioimaging applications. To address this point, we investigated the photophysical properties of representative DAPA fluorophores (*o*-DAPA, *p*-DAPA, **5**, and **8**) in polar solvents, including THF, DMSO, EtOH, and water. The results are summarized in **Response Table 1** on the next page, and included as **Supplementary Table 12** in the revised SI.
- ▶ The fluorescence quantum yield (Φ_F) remains largely insensitive to the solvent polarity, but responds more sensitively to protic solvents. For example, while **5** having strong and well-shielded hydrogen bonds emits efficiently even in water ($\Phi_F = 28\%$), "naked" DAPA fluorophores such as *o*-DAPA or *p*-DAPA, and **8** having solvent-exposed N–H \cdots O hydrogen bond show diminished fluorescence quantum yields in the protic solvents.
- ▶ We postulate that the competitive intermolecular hydrogen bonding with protic solvents could "unfold" the molecule to open up non-radiative decay pathways through bond rotation/vibration that effectively quench the fluorescence emission. This interpretation is in line with "the restriction of internal bond rotations by intramolecular hydrogen bond" (Page 7, Paragraph 2), which we brought up when comparing the Φ_F values of various DAPA fluorophores.

Response Table 1 | Photophysical properties of *o*-DAPA, *p*-DAPA, **5**, and **8** in selected polar solvents.

Compound	Solvent	$\lambda_{\text{abs,max}}$ (nm)	$\lambda_{\text{em,max}}$ (nm)	Stokes shift (cm^{-1})	Φ_{F} (%)
o -DAPA	CHCl ₃	432	531	4320	26
	THF	443	533	3810	29
	DMSO	452	549	3910	26
	EtOH	450	546	3910	6
	H ₂ O	435	572	5510	– ^a
p -DAPA	CHCl ₃	482	618	4570	6
	THF	496	623	4110	5
	DMSO	496	655	4890	3
	EtOH	482	668	5780	– ^a
	H ₂ O	454	693	7600	– ^a
5	CHCl ₃	405	502	4770	73
	THF	404	502	4830	63
	DMSO	371	508	7270	44
	EtOH	386	515	6490	61
	H ₂ O	355	516	8790	28
8	CHCl ₃	445	556	4490	30
	THF	450	556	4240	25
	DMSO	408	554	6460	30
	EtOH	425	572	6050	7
	H ₂ O	395	574	7890	2

^aNegligible fluorescence intensity.

- The diminished fluorescence quantum yields of the "naked" DAPA fluorophores in water would be advantageous in imaging specific organelles having hydrophobic environments, such as lipid droplets (LPs), as we demonstrated previously with other fluorophores ⁴. On the other hand, doubly-functionalized *p*-DAPA, such as **5**, emits efficiently in water (see above), suggesting that bioconjugation with specific organelle-targeting groups might be feasible by straightforward carbonyl substitution reactions. Efforts are currently underway in our laboratory in such directions.

5. Then, why did they select the BH&HLYP functional? This is not explained in neither the manuscript or the SI.

- Unlike the popular TDDFT, the recently developed mixed-reference spin-flip time-dependent density functional theory (MRSF-TDDFT; MRSF for brevity) provides the correct dimensionality of conical intersections between S₁ and S₀. From the previous benchmarking studies ^{5,6}, we found that MRSF requires DFT functionals with a larger fraction of the Hartree–Fock exchange. Thus, MRSF in conjunction with the BH&HLYP density functional shows the best overall agreement with the theoretical best estimates of VEEs, as well as the theoretically available shapes of the ground and excited state PESs and conical intersections between them.
- To make this point clear, the following sentence was added to the Method section of the revised manuscript:

"BH&HLYP functional was employed to provide the best performance of the

⁴ Lee, B. et al. BOIMPY: Fluorescent boron complexes with tunable and environment-responsive light-emitting properties. *Chem. Eur. J.* **22**, 17321–17328 (2016).

⁵ Horbatenko, Y., Lee, S., Filatov, M. & Choi, C. H. How beneficial is the explicit account of doubly-excited configurations in linear response theory? *J. Chem. Theory. Comput.* **17**, 975–984 (2021).

⁶ Lee, S., Shostak, S., Filatov, M. & Choi, C. H. Conical intersections in organic molecules: Benchmarking mixed-reference spin-flip time-dependent DFT (MRSF-TD-DFT) vs spin-flip TD-DFT. *J. Phys. Chem. A* **123**, 6455–6462 (2019).

6. I would like to see NICS values calculated with a larger basis set, including diffuse functions.

- In response to the reviewer’s comment, we have employed larger basis sets to calculate the NICS(1)_{zz} values. As summarized in **Response Table 2** below, the NICS values of *o*-DAPA and *p*-DAPA at four different geometries (S_{0,min}, S₁ FC, S_{1,min}, and S₁@S_{0,min}), used as benchmark systems, are largely insensitive to the choice of basis sets: 6-31G*, 6-31+G*, 6-31++G*, and cc-pVDZ.

Response Table 2 | NICS(1)_{zz} values calculated by CASSCF(2,2) using various basis sets.

	basis set	S _{0,min}	S ₁ FC	S _{1,min}	S ₁ @S _{0,min}
o -DAPA	6-31G*	-23.0	26.6	9.0	-22.4
	6-31+G*	-22.8	28.5	9.1	-22.0
	6-31++G*	-23.6	28.2	8.9	-22.3
	cc-pVDZ	-22.7	27.6	8.8	-22.3
p -DAPA	6-31G*	-24.0	31.2	8.8	-21.6
	6-31+G*	-23.5	30.2	9.1	-20.9
	6-31++G*	-23.5	30.8	9.0	-21.0
	cc-pVDZ	-23.7	30.6	8.1	-21.0

- By extending the tail region of the atomic orbitals, diffuse functions more accurately describe the electrons located remotely from the nuclei. As such, the addition of diffuse functions could improve the quality of calculations on long-range interactions found in negative ions and treatment of diffused electrons in highly excited states.
- For the DAPA systems, however, the addition of diffuse functions for heavy atoms (6-31+G*) or even for hydrogen atoms (6-31++G*) did not significantly affect the NICS values, when a comparison is made with the results from a simpler 6-31G* basis set. We also adopted Dunning’s cc-pVDZ basis set, which includes polarization functions. The trend of NICS values was also commensurate with the results obtained using Pople’s basis sets.
- With no significant long-range interactions in small-sized DAPA molecules, the 6-31G* basis set seems to be sufficient for describing covalent bonds and short-range intramolecular hydrogen bonds, in so far as the NICS values are concerned.

7. With regard to the conical intersections shown in Figure 5 and discussed on page 9, do they correspond to the prefulvenic conical intersection observed for benzene?

- The benzene ring portion of *o*-DAPA adopts a highly distorted geometry at CI₁₀, which appears similar to the structure at the S₁/S₀ prefulvenic conical intersection of benzene. For a more quantitative comparison, we reconstructed the three-dimensional structure of benzene at the conical intersection by using reported Cartesian coordinates (calculated at the SA-CASSCF(6,6)/ANO-RCC-VTZP level)⁷. As shown in **Response Fig. 3** on the next page (also included as **Supplementary Fig. 12** in the revised SI), the C⋯C interatomic distance (2.436 Å) of the puckered 3-membered ring portion of *o*-DAPA is significantly longer than that (1.935 Å) of benzene at CI₁₀. As such, the non-planar

⁷ Slanina, T. et al. Impact of excited-state antiaromaticity relief in a fundamental benzene photoreaction leading to substituted bicyclo[3.1.0]hexenes. *J. Am. Chem. Soc.* **142**, 10942–10954 (2020).

geometry of *o*-DAPA at the conical intersection is best described as twist-boat, rather than genuine prefulvenic conical intersection with the 5-membered ring formed. For *m*-DAPA or *p*-DAPA, the corresponding conical intersections have essentially planar benzene rings, which are even less related to the prefulvenic conical intersection.

Response Fig. 3 | Optimized geometry at CI_{10} of benzene (**a** and **b**), and *o*-DAPA (**c** and **d**) with the $C \cdots C$ interatomic distances of the puckered rings in red.

- We also note that the energy barrier from $S_{1,\min}$ to CI_{10} is 1.06 eV for *o*-DAPA, which is much larger than that (0.37 eV) of benzene⁸. As such, non-radiative relaxation pathways en route to photoisomerization products of benzene, such as fulvene or benzvalene derivatives, are effectively precluded for *o*-DAPA.

8. In the Abstract on page 1 should be “excited-state antiaromaticity” instead of “excited-state aromaticity”

- We thank the reviewer for a careful reading. Corrections were made to clarify the meaning: "A delicate interplay of the excited-state **antiaromaticity** and hydrogen bonding defines the photophysics of this new class of single benzene fluorophores." (Abstract)

9. In my view the subtitle of the section starting at the end of page 10 should be “Excited-State Antiaromaticity as the Origin ...” instead of “Excited State Aromaticity as the Origin ...”.

- The subtitle was revised to read "Excited-state **Antiaromaticity** as the Origin of the Large Stokes Shift”.

⁸ Callomon, J. H., Parkin, J. E. & Lopez-Delgado, R. Non-radiative Relaxation of the Excited \tilde{A}^1B_{2u} State of Benzene. *Chem. Phys. Lett.* **13**, 125–131 (1972).

10. At the end of the first paragraph on page 14 it says “Strokes” instead of “Stokes”

- ▶ This typographical error has been corrected:
"We thus conclude that the large **Stokes** shifts of SBFs originate from ..."

Reviewer #2. We thank the reviewer for a careful and critical reading of our manuscript. He/she stated that *"The manuscript ... is well-written and deals with the highly relevant question of how to make smaller fluorophores with red(-shifted) emission ... In my opinion, this manuscript is original, complete and provides new insights for developing fluorescent organic molecules. I have only minor suggestions for textual improvement listed below. I recommend publication with minor revision."* Listed below are our point-by-point responses to her/his helpful comments and suggestions.

1. I find the title a bit confusing and one word short. If it concerns “the Smallest Red Fluorophore” it would be a lot clearer to me.

- ▶ We agree with the reviewer’s suggestion that the original title might look confusing without the qualifier "Red". To clarify the meaning, the title has been revised to **"Relief of Excited-State Antiaromaticity Enables the Smallest Red Emitter"**.

2. Page 9, line 10:

Here, the S1 state corresponds to a single-electron HOMO → LUMO transition. Wouldn't it be more correct to state that:

Here, the S1 state resulted from a single-electron HOMO → LUMO transition.

- ▶ According to the reviewer’s suggestion, the sentence was revised to read
"Here, the S₁ state **resulted from** a single-electron HOMO → LUMO transition."
(Page 9, Paragraph 2)

3. Page 10, line 9:

Fluorescence spectra were recorded on a Photon Technology International Quanta-Master 400 spectrouorometer with FelixGX software.

Change spectrouorometer to either spectrofluorometer or spectrophotometer

- ▶ The sentence on Page 15, Line 9 has been revised by fixing the typographic error:
"Fluorescence spectra were recorded on a Photon Technology International Quanta-Master 400 **spectrofluorometer** with FelixGX software."

4. In Supplementary Table 3, with entry 1, reference 28 should be mentioned (this one is from the German edition)

- ▶ We thank the reviewer for pointing out this oversight. Correction was made on Page 14 of the revised SI.

Reviewer #3. This reviewer provided a detailed and thoughtful analysis and positive evaluation of our work. He/she recommended publication as it is.

For your reference, an annotated version of the manuscript is provided, in which all the changes/corrections are highlighted (yellow in response to reviewer comments/suggestions; green for editorial changes/corrections requested by your office).

We believe that these changes have satisfactorily addressed all the points raised by the reviewers, and greatly improved the manuscript. We hope that this work can be considered for publication in *Nature Communications*.

Thank you for your efforts and best wishes.

Sincerely,

Dongwhan Lee

DL/d1
Encl.

REVIEWERS' COMMENTS

Reviewer #1 (Remarks to the Author):

I am very happy with the revisions carried out. Yet, I have two points that the authors need to address:

1: On page 9, I think the authors should add one brief sentence telling that the structure of the CI10 does not resemble that of the prefulvenic CI of parent benzene. Just to clarify this to a reader who might wonder.

2: The conclusions on two-electron transfer given in ref. 49 ("Two-electron transfer stabilized by excited-state aromatization") has recently been criticized strongly as computations show very little charge-transfer in the lowest singlet excited state of TMTQ (see Escayola et al in *Angew. Chem. Int. Ed.* 2021, 60, 10225). I.e., the experimental observation on a solvent dependence in the emission can be explained in other ways than by large CT-character of the emissive state. Computations instead reveal that the lowest excited state is stabilized by Hückel-aromaticity, while a state of two-electron transfer character would be located at very high energy according to constrained DFT. Molecules of the TMTQ-type are best described as having lowest excited states of Hückel-Baird hybrid aromatic character with predominant Hückel-aromaticity (see *Chem. Eur J.* 2016, 22, 2793). The authors should take this into account when referring to ref 49 on page 13. At this point, it is important to remark that not all publications claiming excited state Baird-aromaticity are correct.

Reviewer #1. The reviewer commented that "I am very happy with the revisions carried out. Yet, I have two points that the authors need to address." Listed below are our point-by-point responses to her/his helpful comments and suggestions.

1: On page 9, I think the authors should add one brief sentence telling that the structure of the CI10 does not resemble that of the prefulvenic CI of parent benzene. Just to clarify this to a reader who might wonder.

- According to kind suggestion, the following sentence has been added to the revised manuscript:

"The nonplanar geometry at CI₁₀ is best described as twist-boat, rather than genuine prefulvenic CI of benzene^{37,42,43}."
(Page 9, Paragraph 3)

Two articles on the prefulvenic CI of benzene have also been added as references #42 and #43.

42. Palmer, I. J., Ragazos, I. N., Bernardi, F., Olivucci, M. & Robb, M. A. An MC-SCF study of the S₁ and S₂ photochemical reactions of benzene. *J. Am. Chem. Soc.* **115**, 673–682 (1993).

43. Li, Q. et al. A global picture of the S₁/S₀ conical intersection seam of benzene. *Chem. Phys.* **377**, 60–65 (2010).

2: The conclusions on two-electron transfer given in ref. 49 ("Two-electron transfer stabilized by excited-state aromatization") has recently been criticized strongly as computations show very little charge-transfer in the lowest singlet excited state of TMTQ (see Escayola et al in *Angew. Chem. Int. Ed.* 2021, 60, 10225). I.e., the experimental observation on a solvent dependence in the emission can be explained in other ways than by large CT-character of the emissive state. Computations instead reveal that the lowest excited state is stabilized by Hückel-aromaticity, while a state of two-electron transfer character would be located at very high energy according to constrained DFT. Molecules of the TMTQ-type are best described as having lowest excited states of Hückel-Baird hybrid aromatic character with predominant Hückel-aromaticity (see *Chem. Eur J.* 2016, 22, 2793). The authors should take this into account when referring to ref 49 on page 13. At this point, it is important to remark that not all publications claiming excited state Baird-aromaticity are correct.

- We thank the reviewer for bringing this recent publication in *Angew. Chem. Int. Ed.* to our attention, which challenges the initial claims made in *Nat. Commun.* (2019), cited as reference #49 in the previous version of the manuscript. Apparently, the involvement of CT-type states (accessed by two-electron transfer in the excited-state) in the emission is yet to be settled on. Moreover, regardless of its validity, such a mechanism has little conceptual relevance to the relief of ESAA assisted by hydrogen bonding, which is the key discovery of our work. To avoid unintended confusion and digression, we decided to remove the previous reference #49. The revised sentence now reads:

"Peripheral functional groups mediate proton transfer (PT)^{49,50} or proton-coupled electron transfer (PCET)⁵¹ to relieve the antiaromaticity of π -systems in the excited-state."
(Page 13, Paragraph 3)

The subsequent references have been renumbered accordingly.